# A Rapid Screening Platform for Simultaneous Evaluation of Biodegradation and Therapeutic Release of an Ocular Hydrogel

**DOI:** 10.3390/pharmaceutics15112625

**Published:** 2023-11-15

**Authors:** Brandon Ho, Chau-Minh Phan, Piyush Garg, Parvin Shokrollahi, Lyndon Jones

**Affiliations:** 1Centre for Ocular Research & Education (CORE), School of Optometry & Vision Science, University of Waterloo, 200 University Avenue West, Waterloo, ON N2L 3G1, Canada; brandon.ho@uwaterloo.ca (B.H.); piyush.garg@uwaterloo.ca (P.G.); p.shokrollahi@uwaterloo.ca (P.S.); lyndon.jones@uwaterloo.ca (L.J.); 2Centre for Eye and Vision Research (CEVR), 17W Hong Kong Science Park, Hong Kong

**Keywords:** millifluidics, image analysis, biodegradation, hydrogels, ocular drug delivery

## Abstract

This study attempts to address the challenge of accurately measuring the degradation of biodegradable hydrogels, which are frequently employed in drug delivery for controlled and sustained release. The traditional method utilizes a mass-loss approach, which is cumbersome and time consuming. The aim of this study was to develop an innovative screening platform using a millifluidic device coupled with automated image analysis to measure the degradation of Gelatin methacrylate (GelMA) and the subsequent release of an entrapped wetting agent, polyvinyl alcohol (PVA). Gel samples were placed within circular wells on a custom millifluidic chip and stained with a red dye for enhanced visualization. A camera module captured time-lapse images of the gels throughout their degradation. An image-analysis algorithm was used to translate the image data into degradation rates. Simultaneously, the eluate from the chip was collected to quantify the amount of GelMA degraded and PVA released at various time points. The visual method was validated by comparing it with the mass-loss approach (R = 0.91), as well as the amount of GelMA eluted (R = 0.97). The degradation of the GelMA gels was also facilitated with matrix metalloproteinases 9. Notably, as the gels degraded, there was an increase in the amount of PVA released. Overall, these results support the use of the screening platform to assess hydrogel degradation and the subsequent release of entrapped therapeutic compounds.

## 1. Introduction

Despite advancements in ocular drug delivery, the eye’s complex physiological barriers continue to pose challenges for sustained and controlled drug release. Drugs and therapeutics administered topically have very low bioavailability due to tear-fluid turnover, nonspecific absorption and blinking [1,2,3]. As a result, novel therapeutic approaches for topical applications such as nanoparticles, liposomes, gels and novel hydrogels are being developed. These technologies aim to increase the drug retention time, overcome ocular barriers and improve bioavailability [4,5,6,7,8].

Among the novel drug-delivery methods are biodegradable hydrogels, which are designed to release entrapped drugs or therapeutics as the gel degrades in situ. Studies have demonstrated that drugs released from degradable materials can be controlled and sustained over several days [9,10,11,12]. In comparison to other drug-delivery approaches, the release of therapeutics from degradable materials, assuming that the therapeutic is entrapped, largely depends on the degradation rate rather than just passive diffusion [9]. Consequently, it is possible to achieve drug-release kinetics by using biodegradable hydrogels that approach the ideal zero-order release [11,12].

The primary challenge in developing biodegradable hydrogels for drug delivery lies in accurately measuring their degradation over time, while also simultaneously being able to evaluate the drug-release kinetics. Standard approaches to assess the biodegradation of hydrogels are either by volume or mass loss, the latter of which is more common [13,14,15]. When assessing mass loss, researchers measure the weight of the dried gel at a specified time point [14,15]. This procedure is exceptionally cumbersome and time consuming because a dried gel from one time point cannot be reused for subsequent measurements. Moreover, the generated dataset would contain significant variability because the same gels cannot be assessed repeatedly over time. Furthermore, the mass-loss method of measuring degradation would not allow researchers to simultaneously monitor drug release.

To properly evaluate the biodegradation of an ocular device or its drug-release kinetics, it is essential to also simulate key factors of the ocular environment, such as the ocular temperature, tear flow and low tear volume. However, most drug-release kinetics for ocular drug delivery are typically performed in a static vial at room temperature [16,17]. This setup may not simulate the eye’s dynamic biological conditions, especially the low tear volume or flow. Consequently, the results obtained by using these simple models might not reflect the release kinetics or biodegradation profile expected in the actual in vivo situation [18].

Millifluidic or microfluidic devices can potentially offer a more representative simulation environment of the eye by mimicking low tear volume and flow. These devices have been used in lab-on-a-chip and even organ-on-a-chip systems to emulate various human organs [19,20]. Notably, these technologies have already been implemented in drug-discovery and -delivery applications [21,22,23,24]. Beyond a more accurate biological simulation, these devices also significantly reduce reagent consumption while providing an increased assay throughput. The main barrier to adopting these devices for research has been their high cost. However, with the advancements in new fabrication technologies such as 3D printing and laser lithography, producing these devices, especially for customized chip and tailored assays, has become more cost effective [25,26].

Thus, the application of millifluidic and microfluidic devices could help develop a screening system to measure the degradation rate of biodegradable hydrogels and the subsequent drug or therapeutic release. Gelatin methacrylate (GelMA) was chosen as the biodegradable hydrogel in this study due to its broad applications in drug delivery [27,28]. Its biodegradation can be accelerated by matrix metalloproteinases (MMPs), the enzymes present in the tear film [29,30]. The degradation rate of GelMA can be controlled by modulating the levels of MMPs to which the gel is exposed. Polyvinyl alcohol (PVA) was selected as the model therapeutic agent contained within the biodegradable gel given its large molecular size, which ensures its entrapment within the gel matrix. Additionally, PVA is also used commonly as an ocular lubricant [31,32]. The aim of this study was to develop a screening platform by using a custom millifluidic device coupled with automated imaging analysis to simultaneously monitor both GelMA degradation and the subsequent release of PVA.

## 2. Materials and Methods

### 2.1. Preparation of GelMA

GelMA was prepared according to previously reported methods [33]. In brief, type A gelatin from porcine skin (10% *w*/*v*, Sigma-Aldrich, St. Louis, MO, USA) was reacted with methacrylic anhydride (1% *v*/*v*, Sigma-Aldrich, St. Louis, MO, USA) in a carbonate bicarbonate (CB) buffer (12.5 mM sodium bicarbonate, 87.5 mM sodium carbonate anhydrous, pH 9.4) at 50 °C for 1 h. This substitution reaction was stopped by adding acetic acid to a final concentration of 0.15% (*v*/*v*). The resulting product was dialyzed in 12–14 kDa dialysis tubes (Sigma-Aldrich, St. Louis, MO, USA) in deionized water for 24 h. Following the dialysis step, the solution was freeze-dried and then stored at −80 °C until use.

### 2.2. Preparation of GelMA-PVA Hydrogels

The freeze-dried GelMA was reconstituted in a solution of PBS to 10% *w*/*v* and then UV crosslinked by using a photoinitiator (lithium phenyl-2,4,6-trimethylbenzoylphosphinate (LAP)). The GelMA hydrogels were cast into a disc shape with a radius of 2 mm and a thickness of 1 mm. A similar method was used to fabricate the GelMA-PVA hydrogels, which contained a 7.5% concentration of PVA. GelMA (20% *w*/*v*) was added to a LAP (1.2% *w*/*v*) solution prepared in PBS. A PVA (15% *w*/*v*) solution was prepared in PBS and heated for complete dissolution (~37 °C). The two solutions were added in equal amounts to make the GelMA/PVA prepolymer solution, which comprised GelMA (10%), PVA (7.5%) and LAP (0.6%). This prepolymer solution was then used to make polymer discs.

### 2.3. Biodegradation of GelMA with MMP9 by Mass Loss in a Vial

The GelMA hydrogel discs were dried by briefly dabbing the biomaterial with lens paper, and their initial weights were measured on a scale and recorded for t = 0. The hydrogel discs were then added to 1.7 mL microcentrifuge tubes which contained 1 mL of varying MMP9 (Gibco, Billings, MT, USA) concentrations (0, 25, 50, 100 and 200 μg/mL) in 1× phosphate buffered saline (PBS). The samples were incubated at 37 °C with gentle agitation. At t = 0, 4, 8, 12, 16 and 24 h, the samples were removed from the solution, dried and the dried weight was recorded.

### 2.4. Biodegradation of GelMA with MMP9 Using a Custom-Designed Millifluidic Device System

A custom-designed polydimethyl siloxane (PDMS) millifluidic device was provided by EyesoBio Inc. (Waterloo, ON, Canada). The device contains a series of microfluidic channels (2 mm width and 50 mm length) with a circular chamber (5 mm radius and 2 mm height) at the center for placing the samples (refer to Figure 1). Since GelMA is relatively transparent, the GelMA hydrogel discs were stained with a red dye solution (0.1% *v*/*v*, ClubHouse red food coloring, McCormick & Company, Baltimore, MD, USA) in 1× PBS to help visualize the degradation process. The entire device was connected by using Teflon tubing (1/16” inner diameter × 1/32” wall Tygon Tubing, Saint Gobain Performance Plastics, Courbevoie, France) with the inlet tubes connected to a 20 mL glass source-media reservoir (WHEATON^®^ vials, Sigma-Aldrich DWK986541, St. Louis, MO, USA). The outlet tubes were connected to a peristaltic pump (Darwin Microfluidics BT100-1L, Paris, France). An overview of the experimental setup is shown in Figure 1.

Different concentrations of the MMP9 enzyme were prepared in 1× PBS (0, 50, 100 and 200 μg/mL) with 0.1% (*v*/*v*) red dye and were added to the source-media reservoirs. The entire experimental setup was placed inside a custom-designed acrylic chamber with temperature and humidity control equilibrated to 37 °C for 30 min prior to starting the biodegradation time course. Once equilibrated, the MMP9 solution flowed through the millifluidic device at 300 μL per minute. For comparison, using a different set of samples, the mass loss was also measured for the GelMA discs treated with the different concentrations of MMP9.

### 2.5. Image Analysis of Hydrogel Degradation

Images of the hydrogel biodegradation were obtained by using an iPhone 12 (Apple, Cupertino, CA, USA) running a time-lapse application (Life Lapse, Vancouver, BC, Canada), acquiring images every 60 s for the entirety of the time course (t = 20 h). Automated computational quantification of the hydrogel biodegradation was performed by using custom-written Python code (Python v3.11.3, https://www.python.org/ (accessed on 21 July 2023)). Images at each time frame were opened, and pixels in the entire image were scanned for a specific range of red color, corresponding to the hydrogel stained by the red dye. An image mask was created, such that those pixels identified as ‘red’ were assigned a value of 1, and all the others were assigned a value of 0. Next, the analysis pipeline distinguished between different hydrogel discs through grouping the image pixels that had a value of 1 and were near one another. Finally, the pipeline counted the number of pixels for each group (each hydrogel insert), and this was plotted over time. The values for the percentage (%) change in the GelMA hydrogel discs were calculated by dividing the surface area of the GelMA (in pixels) determined at each time point by the initial surface area at the start of the experiment. All the plots related to the time-lapse quantification of hydrogel biodegradation were generated by using Python and the Seaborn library to make statistical graphics in Python.

### 2.6. Estimation of Average Biodegradation Rates

To estimate the average biodegradation rates of the hydrogel discs, the initial and final measurements of the hydrogel surface area (in counted pixels), or the time at which the insert fully degraded if the insert was no longer present by the end of the time course, was recorded. An assumption was made that the hydrogel degrades linearly to compare the approximated rates of degradation between different samples. The change in the surface area was normalized to the initial surface area of the hydrogel and divided by the time of the experiment, or the time to reach complete degradation, to obtain average rate measurements of the percent change in the hydrogel material per hour (%/h).

### 2.7. Release of PVA from Millifluidic Chip

All the GelMA-PVA hydrogel discs (10% GelMA, 7.5% PVA) analyzed in this study were prepared and used within 24 h. GelMA-PVA discs were placed in the circular chambers of the custom millifluidic device and treated with an MMP9 solution (0 and 200 μg/mL) using the same custom-designed experimental system described above. At t = 0, 1, 2, 3 and 17 h, 50 μL of the solution was removed from the glass reservoir and stored at 4 °C until the end of the experiment. Each sample was then used for the detection of PVA.

### 2.8. Detection of PVA

The detection of PVA was achieved by using previously published protocols [34,35]. In brief, a PVA-detection solution was formulated by using a solution of iodine (150 mM potassium iodide, 50 mM diiodine in deionized water, Sigma-Aldrich 221945 and 207772, St. Louis, MO, USA) and borate (64.7 mM boric acid in deionized water, Sigma-Aldrich B0394, St. Louis, MO, USA). To measure PVA in the solution, 150 μL of the PVA-detection solution was added to 50 μL of the sample in a 96-well plate and incubated at room temperature for 20 min with gentle shaking. The absorbance of the unknown samples, in addition to a set of standard samples of known PVA concentrations, prepared in PBS was then measured at 630 nm by using an Imaging Multimode Plate Reader (Cytation 5, Agilent BioTek Instruments, Winooski, VT, USA).

### 2.9. Quantification and Statistical Analysis

Statistical analyses were performed by using GraphPad Prism version 9.4.0 (GraphPad, La Jolla, CA, USA) and Python v3.11.3. To conduct significance tests for multiple comparisons of groups, a one-way analysis of variance (ANOVA) was used, as described in the figure legends. For linear correlations to compare the degradation profile between a conventional mass loss and the visual technique, the Pearson R correlation coefficient was used. Results with *p* < 0.05 were considered statistically significant: * *p* < 0.05, ** *p* < 0.005.

## 3. Results

### 3.1. Combined Millifluidics Imaging and Computational Analysis for Quantitative Characterization of GelMA Biodegradation

The work described here involves an experimental setup coupling a millifluidic device with a computational image-analysis pipeline that quantitatively measured the biodegradation of GelMA hydrogel discs continuously over the course of 20 h (Figure 2a). A series of time-lapse images of the GelMA biodegradation was acquired, and an image-analysis pipeline was applied (Figure 2b), which identified the image pixels that corresponded only to the GelMA material and recorded the change in the pixel area of the GelMA over time. As expected, the GelMA insert visibly degraded over time with MMP9 exposure, with complete biodegradation taking over 10 h when treated with 100 μg/mL MMP9, as seen in Figure 2c.

Two important observations were noted when utilizing the millifluidic chip. Firstly, the time to degrade GelMA was longer in these millifluidic devices compared to the GelMA degraded in a static 1 mL MMP9 solution (100 μg/mL) at the same concentration. The bulk degradation of GelMA in the static condition, with gentle agitation (50 rpm) at 37 °C, increased with greater concentrations of MMP9 and completely degraded around 8 h after treatment with the 100 μg/mL MMP9 solution (Figure 2d). In contrast, the complete degradation of the gels in the millifluidic chip took approximately 12 h in the 100 μg/mL MMP9 solution (Figure 2c). Secondly, the decrease in the GelMA disc dimensions during the hydrogel biodegradation within the device was more pronounced in the disc radius rather than the thickness (Figure 2e). This contrasts with the GelMA discs degraded by 100 μg/mL of MMP9 in a conventional microfuge tube, whereby the GelMA object is surrounded by the MMP9 solution, and an apparent reduction in the GelMA disc thickness can be observed in Figure 2f. It should be noted that the thickness of the discs within the device could not be observed during the course experiment, as images were acquired by using a top-down approach.

Next, the ability of this system to accurately discern differences in the GelMA biodegradation rates was tested. The custom-designed millifluidic device could simultaneously accommodate eight samples; it provided a constant flow of seven different concentrations of the MMP9 solution and PBS as a control (Figure 3). The computational image-analysis pipeline was applied to these treated samples, and the change in the GelMA surface area over time was quantified (Figure 3a,b). There was a significant increase in the biodegradation rates with greater concentrations of MMP9 (Figure 3c–e). A similar trend was also observed when the mass-loss method was used to measure degradation for the GelMA discs in a static solution (Figure 2d).

It should be noted that very high MMP9 concentrations (>100 μg/mL) produced results with considerable variation between technical replicates (Figure 3c). In fact, at very high concentrations (200 μg/mL), the degradation was rapid and irregular. In some instances, a drastic decrease in the detectable material was observed for the GelMA, suggesting a sudden disintegration of the material rather than a gradual reduction in size that was observed in the GelMA hydrogel. Indeed, visible biodegradation debris could be seen flowing out of the device (Figure 3d). Despite these sources of variation, the overall average rates of GelMA degradation across the entire experiment generally agreed between biological replicates (Figure 3e). Most importantly, there was a very high positive agreement between the computationally measured sizes of the GelMA hydrogel samples as compared to their mass loss (R = 0.91, Figure 3f).

### 3.2. Release of Model Therapeutic PVA Can Be Quantified from the Millifluidic Device

Next, the simultaneous biodegradation and release of an encapsulated molecule were evaluated from the GelMA hydrogel discs. To this end, the release of PVA (as a model therapeutic) was quantified from the eluates from the millifluidic device, as illustrated in Figure 4. It was observed that the degradation of the GelMA-PVA hydrogel was substantially faster than that of the GelMA hydrogels alone (Figure 4a). Nevertheless, there was a detectable sustained release of PVA from the GelMA-PVA hydrogels over the 17 h of the MMP9 treatment (Figure 4b). Interestingly, PVA was detected even in the absence of MMP9, which suggests that some PVA can still be released from the hydrogel via passive diffusion. Indeed, a burst of PVA was detected at 1 hr of treatment in both enzyme and without-enzyme conditions. However, there was a greater increase in PVA detected over time in the eluates in the presence of MMP9 when compared with GelMA-PVA (Figure 4b). In addition to monitoring PVA release from the system, degraded GelMA was also quantified since it would also be present in the eluates. A high correlation (Pearson R = 0.97) was observed between the measured GelMA material in the eluates compared to the apparent hydrogel disc size quantified by the visual method (Figure 4c).

## 4. Discussion

Quantifying hydrogel degradation is of significant interest in the drug-delivery field as the degradation profile of a biodegradable material can be leveraged for controlled drug delivery [10,11,12,36,37]. However, current methods to evaluate the biodegradation of biomaterials are extremely tedious and prone to significant experimental variability. This is because the samples must be manually removed from the experimental system at each time point, dried and then measured. Consequently, researchers can choose only a feasible number of time points to capture snapshots of the degradation profile. However, for some material formulations, there are moments during degradation when the material can disintegrate rapidly over a short span (as this study noted), especially near the end of the degradation profile. Such crucial moments might be missed with just a snapshot of the degradation profile. Thus, novel in vitro assays to better characterize hydrogel and/or biomaterial degradation kinetics are required.

### 4.1. Novel Quantitative Millifluidic Imaging System to Quantify Hydrogel Biodegradation

This study aimed to develop a high-throughput screening methodology to continuously monitor the biodegradation of a GelMA hydrogel. The system employs a millifluidic chip to provide a steady flow, which can be adjusted as necessary to replicate physiological conditions or simulate accelerated degradation conditions. The system in this study was set up to run a total of 24 samples simultaneously, but this capacity can easily be expanded as required by either adding more sample channels to the device or connecting multiple devices in a series. The device also utilizes an automated imaging algorithm to acquire and analyze thousands of images of the degradation process over time without any need for manual measurements. Importantly, the computationally calculated degradation profile obtained by using our automated device and imaging platform correlated highly compared to the conventional mass-loss approach (Pearson R = 0.91). These results support the use of this visual-analysis method as an orthogonal approach to measure material degradation.

Despite the numerous advantages this automated system has over the more conventional mass-measurement approach, several considerations should be noted. Firstly, the current system only measures the surface area from a top view. Thus, thin samples, ideally in a disc shape, with a large front and back surface area, would be best suited for this analysis. Since this system does not measure material thickness, degradation measurements may deviate from the mass-loss method as the thickness of the material increases. Secondly, this method requires that the hydrogel exhibits a strong visual contrast against the background. This is unlikely for many hydrogels, such as GelMA, as they are transparent. Therefore, a hydrophilic red food dye was used to both stain the GelMA gels and was added to the MMP9 solution to outline the gel’s overall shape as it degraded. The assumption is that the dye’s absorption into the sample does not influence the sample’s degradation rate. However, this assumption might not always be valid, as some chemical compounds can affect material properties [38,39]. Hence, it is important that the dye selected for this screening should have minimal impact on the properties of the hydrogel. Alternatively, if the material is fluorescent, then capturing images via fluorescence imaging is also possible with this approach. Nevertheless, the selection of an appropriately shaped biomaterial that can be stained with an inert dye would be highly suitable in this system.

One major advantage of this millifluidic system is that it allows for the collection of eluates for subsequent assays, enabling the simultaneous monitoring of biodegradation and the release of a therapeutic from the same sample. In this study, both the levels of GelMA and PVA present in the eluate were measured to (1) provide an additional method to monitor GelMA degradation and (2) evaluate the release of a loaded model therapeutic from a hydrogel. The degradation of GelMA was consistent over time and exhibited a strong linear correlation with the visual-analysis method (Pearson R= 0.97). Used in combination with the visual-analysis method, this assay aids in producing a reliable degradation profile. Furthermore, drugs that have been released from hydrogels may be used in conjunction with mathematical modeling to determine the type of release mechanisms that may be exhibited. In the conditions of this study, MMP9 likely acts by breaking down long polymeric chains of GelMA into smaller chains, causing the faster release of entrapped PVA from the inner core of the biomaterial. This mechanism could be described by Hopfenberg’s model, but the true release kinetics will depend on the specific type of degradable polymer used, geometry and shape of the matrix and the presence of other chemical agents, among other factors [40]. In future studies, eluates from this system can be subjected to more sophisticated modeling of drug-release kinetic profiles as well as toxicity testing with cells. For example, eluted therapeutics with a well-defined release profile from this millifluidic system could be directly applied to human corneal epithelial cells exposed in a control and diseased state, such as hyperosmotic conditions to mimic dry eye syndrome. Cellular phenotypes, such as viability, the expression of proteins of interest and cellular morphological changes could be assessed through conventional means or through live-cell microscopy.

### 4.2. Potential for High-Throughput Screening of Hydrogels Used for Drug Delivery

One notable observation made in this study was that the addition of PVA into the GelMA disc could alter its biodegradation kinetics. Unsurprisingly, differences in the chemical composition of a hydrogel are one of many factors that can influence the material’s physical and chemical properties, which ultimately influence the release kinetics of an encapsulated drug [41]. Other parameters that must be considered include the flow rates, volume, pH and temperature of the experimental system.

In the context of the ocular surface, there are other factors, such as blinking, that could affect the degradation rate. For instance, as the gel degrades, its internal structures may weaken, and even a small force from blinking could cause the gel to disintegrate instantly. For these reasons, a high-throughput and automated experimental system, such as the one presented in this study, would be beneficial for testing all these conditions and parameters simultaneously. Furthermore, recent technological advances have drastically increased the number of unique hydrogels that can be synthesized at one time [42,43]. One limiting factor could be the characterization of all these different formulations. The system developed in this study could significantly aid in the screening of these formulations. Finally, given that most degradation-time-course studies for polymers can span days, weeks or even months [44], an imaging system that allows researchers to quantitatively monitor the degradation remotely would reduce the time required to identify promising candidate materials. In the future, the goal is to also incorporate real-time analysis into this system as image data are gathered. Furthermore, it would be valuable to conduct in vitro–in vivo correlation studies in future studies to calibrate the system’s parameters and generate representative physiological degradation profiles.

## 5. Conclusions

The developed screening platform was able to automatically monitor the degradation of a GelMA hydrogel by using a visual-imaging algorithm. The visual analysis showed a very strong positive correlation with the conventional mass-loss method used to measure material degradation. Furthermore, the developed system can also be used to measure the release of biodegraded material or a therapeutic from the same sample. In the future, this type of screening system could be adapted to measure the dissolution of various materials.

## Figures and Tables

**Figure 1 pharmaceutics-15-02625-f001:**
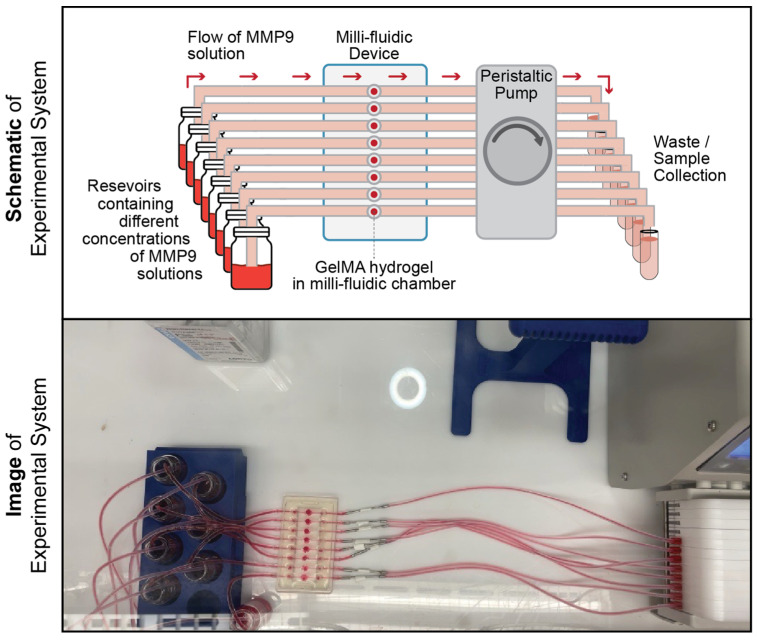
Setup of the millifluidic system for analyzing the biodegradation of GelMA hydrogels. A schematic diagram and photograph from a top-down view of the millifluidic system used in this study. The bottom photograph of the setup was acquired from the same fixed camera module used for all image-based biodegradation experiments. The depicted photo is thus representative of the quality (color accuracy and image resolution) of all time-lapse images in all biodegradation experiments.

**Figure 2 pharmaceutics-15-02625-f002:**
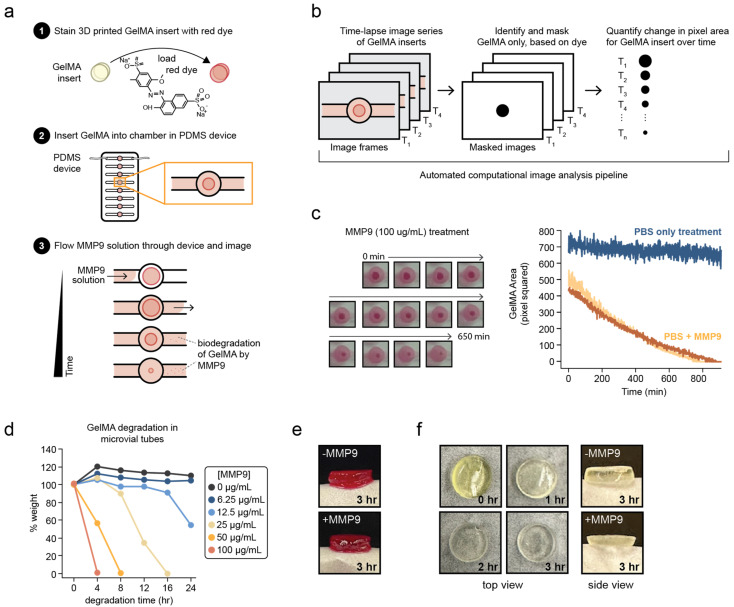
Time-lapse millifluidic imaging system with computational image-analysis capture kinetics of MMP9-dependent GelMA degradation. (**a**) Schematic diagram of the experimental approach to visually observe GelMA biodegradation. (**b**) Overview of the computational pipeline applied to all images acquired in the time-lapse experiments. (**c**) Representative images of stained GelMA discs within the millifluidic chamber over the course of 650 min of 100 μg/mL MMP9 treatment acquired via time-lapse imaging. Quantification of the change in GelMA disc size over time for discs with or without 100 μg/mL MMP9 in PBS (light and dark orange, and blue line traces, respectively, n = 2). (**d**) Percent of initial GelMA disc weight is plotted over time after exposure to the indicated concentrations of MMP9 in PBS. (**e**) Side view of a GelMA ocular disc after treatment with 100 μg/mL MMP9 in the millifluidic device. (**f**) Representative images (both top-down and side views) of GelMA ocular disc before and after treatment with 100 μg/mL MMP9 in a static vial.

**Figure 3 pharmaceutics-15-02625-f003:**
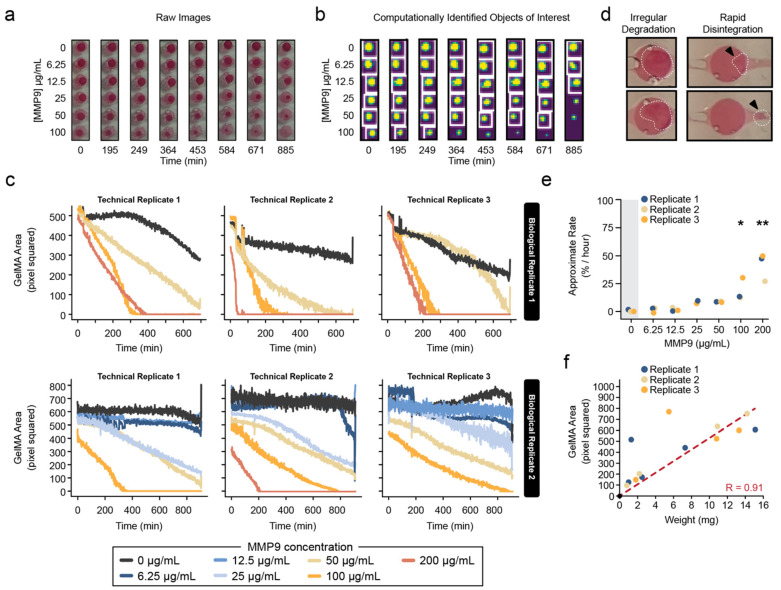
Computational image-analysis pipeline detects differences in GelMA biodegradation, which correlates well with conventional weight-based measurements of hydrogel biodegradation. (**a**) Representative images of GelMA inserts in custom-designed 8-chamber millifluidic devices during treatment with MMP9 over the course of 20 h. (**b**) Application of our automated custom image-analysis pipeline. Yellow indicates areas detected as stained GelMA inserts, and purple denotes areas not labeled as GelMA. (**c**) Quantification of two biological replicates (each with three technical replicates) of MMP9-dependent degradation of GelMA discs using our automated image pipeline. (**d**) Representative images of GelMA hydrogels undergoing irregular degradation or rapid disintegration during treatment with 200 μg/mL MMP9. White dotted outline indicates the location of the hydrogel, and black arrows indicate the location of GelMA biomaterial debris. (**e**) Pseudorates of GelMA biodegradation were calculated and plotted against MMP9 concentration (n = 3). Statistical significance determined by one-way ANOVA with Tukey multiple comparisons test: * *p* < 0.05, ** *p* < 0.005 (**f**) Linear correlation between the size of the GelMA discs and the weight of the same discs after being removed from the millifluidic device. Red dashed line represents the line of best fit with the Pearson correlation coefficient indicated (R).

**Figure 4 pharmaceutics-15-02625-f004:**
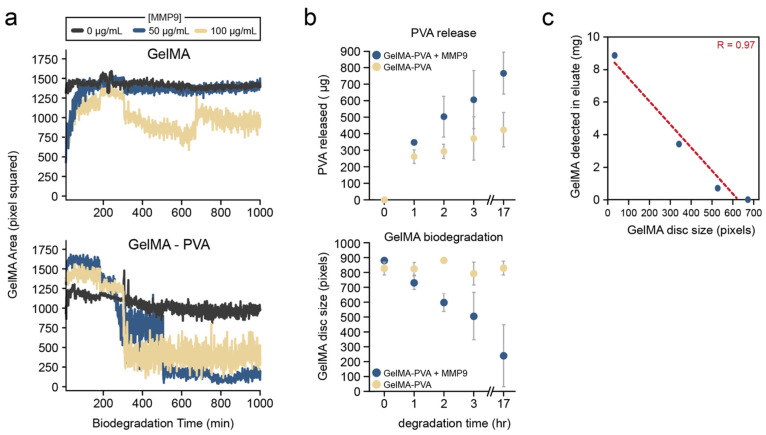
Simultaneous measurements of hydrogel degradation and concentrations of released compounds in device eluates. Computational quantification of the biodegradation of (**a**) GelMA and GelMA-PVA discs MMP9 at 0, 50 and 100 μg/mL. (**b**) Quantification of soluble PVA released from GelMA-PVA discs (top) and detection of degraded GelMA polymer (bottom) at the indicated times following MMP9 treatment. Error bars represent standard error of the mean. (**c**) Correlation between detected GelMA in the eluates and the size of the GelMA-PVA disc in the millifluidic device. Red dashed line represents the line of best fit with the Pearson correlation coefficient indicated (R).

## Data Availability

The data can be shared upon request.

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
