# Peer review of "A Rapid Screening Platform for Simultaneous Evaluation of Biodegradation and Therapeutic Release of an Ocular Hydrogel"

_pharmaceutics, 2023, doi:10.3390/pharmaceutics15112625_

Round 1
Reviewer 1 Report
Comments and Suggestions for Authors
"The paper by Ho and coauthors presents an innovative approach utilizing a millifluidic device coupled with automated image analysis to assess Gelatin methacrylate (GelMa) degradation and the subsequent release of an entrapped wetting agent, polyvinyl alcohol (PVA). This research holds substantial promise in its field, displaying commendable organization in the manuscript. However, several key areas warrant attention before considering this work for publication:
1. Figures are not seamlessly integrated into the manuscript, affecting the overall readability. A thorough review and correction of these figures are essential.
2. What is the quantitative relationship between computationally measured sizes for GelMA hydrogel samples and their corresponding mass loss?
3. What is the degradation rate for the samples?
4. For deeper insights into the mechanisms governing release (erosion, erosion + diffusion, or others), authors should evaluate generalized mathematical models in the release profiles (https://doi.org/10.3390/pr10061094).
Focusing on these areas will significantly enhance the manuscript's completeness and strengthen its potential for publication.
Reviewer 2 Report
Comments and Suggestions for Authors
The research work “A rapid screening platform for simultaneous evaluation of biodegradation and therapeutic release of an ocular hydrogel” aims to address the challenge of accurately measuring the degradation of biodegradable hydrogels that affect drug release. The research team proposed an innovative screening platform using a millifluidic device. This device coupled with automated image analysis that can measure the degradation of Gelatin methacrylate and release of an entrapped wetting agent, polyvinyl alcohol. The work looks novel and promising, however accuracy and flexibility need to prove.
The comments are as follows.
1. Is there any other technique available? If yes, how this system is different. Include this in introduction.
2. Line 102: heated for complete dissolution, mention temperature used.
3. Line 97: lithium phenyl-2,4,6-trimethylbenzoylphosphinate (LAP). What is need? It’s not discussed in introduction and abstract at all.
4. Is it safe to used lithium phenyl-2,4,6-trimethylbenzoylphosphinate for eye?
5. Figure 2 is not clearly visible in pdf. Need to rearrange. (some part cut-off)
6. How much accurate this device is?
Reviewer 3 Report
Comments and Suggestions for Authors
Review of pharmaceutics-2719225
After careful examination of the submission pharmaceutics-2719225 entitled “A rapid screening platform for simultaneous evaluation of biodegradation and therapeutic release of an ocular hydrogel”, I conclude:
1. The submission is devoted to very important issue, conclusions correspond to the collected data, and employed methods and equipment are up to date.
2. Everything is written in grammatically correct English, but I’d rather recommend to take a second look since some statements look strange, e.g.: r.93 Following dialysis, the solution was freeze-dried and stored at -80°C until further use.
3. As about Figures 2-4, the text and images are too compressed. I will definitely recommend arrange panels vertically, that allows to make images larger.
4. On all figures I recommend to make the text BLACK, not grey(gray).
5. It looks strange, that chapters and sections are not numbered according to MDPI style (see e.g. https://doi.org/10.3390/pharmaceutics15112576):
Therefore, I recommend to correct all these discrepancies before acceptance.
Comments on the Quality of English Language1. Everything is written in grammatically correct English, but I’d rather recommend to take a second look since some statements look strange, e.g.: r.93 Following dialysis, the solution was freeze-dried and stored at -80°C until further use.
Round 2
Reviewer 1 Report
Comments and Suggestions for Authors
In my opinion, the new version of the manuscript is suitable for publication.